# Exploring Multi-decadal Time Series of Temperature Extremes in Australian Coastal Waters

Michael Paul Hemming[1], Moninya Roughan[1], and Amandine Schaeffer[1,2]

[1]Coastal and Regional Oceanography Lab, Centre for Marine Science and Innovation, UNSW Sydney, Sydney, NSW 2052 Australia
[2]School of Mathematics and Statistics, UNSW Sydney, Sydney, NSW 2052 Australia.

**Correspondence:** Michael Paul Hemming (m.hemming@unsw.edu.au)

**Abstract.**

The intensity and frequency of extreme ocean temperature events, such as Marine Heatwaves (MHWs) and Marine cold-spells (MCSs), are expected to change as our oceans warm. Little is known about marine extremes in Australian coastal waters, particularly below the surface. Here we introduce a multi-decadal observational record of extreme ocean temperature events starting in the 1940s/50s between the surface and the bottom (50-100m) at 4 long-term coastal sites around Australia; the Australian Multi-decadal Ocean Time Series EXTreme (AMDOT-EXT) data products. The data products include indices indicating the timing of extreme warm and cold temperature events, their intensity, and the corresponding temperature time series and climatology thresholds. We include MHWs, MCSs and shorter duration heat spikes and cold spikes. For MHWs and MCSs, which are defined as anomalies above the daily-varying $90^{th}$ and $10^{th}$ percentile, respectively, and lasting more than 5 days, we also provide further event information, such as their category, and onset and decline rates. The 4 data products are provided as CF-compliant NetCDF files and it is the intention that they be updated periodically. It is advised that data users seek the latest data product version using the DOI in the Code and Data Availability Section. Using these multi-decadal data products, we show the most intense and longest extreme temperature events at these sites, which have occurred below the surface. These data records highlight the value of long-term full water column ocean data for the identification of extreme temperature events below the surface.

## 1 Introduction

Long-term ocean temperature records have become extremely valuable as the oceans warm due to anthropogenic climate change (Masson-Delmotte et al., 2021). This warming is expected to affect the characteristics of extreme temperature events such as Marine Heatwaves (MHWs) and Marine cold-spells (MCSs) (Oliver et al., 2018; Frölicher et al., 2018; Schlegel et al., 2021) (which are often defined to last 5 days or more) and their shorter variants lasting less than 5 days referred to as heat spikes (HSs) and cold spikes (CSs) (Hobday et al., 2016; Schlegel et al., 2021). Extreme warm and cold water events are having an impact on marine ecosystems, including coral bleaching (Zapata et al., 2011; Hughes et al., 2017), mass mortality of organisms (Woodhead, 1964; Garrabou et al., 2009; Santos et al., 2016), and shifts in the distribution of marine species (Vergés et al., 2014; Firth et al., 2015; Wernberg et al., 2016; Smale et al., 2019). The health of a marine ecosystem influences

the amount of life that it can sustain, and thus MHWs and MCSs can affect fisheries and the blue economy (Smith et al., 2021). Globally, MHWs are expected to become more intense, more frequent, and longer-lasting as a result of climate change (Oliver et al., 2018; Frölicher et al., 2018), while the opposite is expected for MCSs (Schlegel et al., 2021). Similarly, in waters off southeastern Australia, there has been an upward trend in surface MHW frequency, intensity, and duration between 1982 and 2020 (Kajtar et al., 2021).

There are numerous examples of extremely warm or cold temperatures being recorded in Australian coastal waters. In 2011 off Western Australia, low sea level pressure linked to a strong La Niña intensified the Leeuwin current leading to MHW conditions at the surface. This MHW lasted for weeks and maximum intensities peaked at 5°C warmer than the 2000–2009 average climatology (Feng et al., 2013). In 2015 / 2016 off eastern Tasmania, the East Australian Current (EAC) extension waters were warmer than average leading to MHW conditions. The MHW at the surface persisted for 251 days between September 2015 and May 2016 and regionally-averaged sea surface temperature anomalies were between 1.5 and 3°C warmer than average between November 2015 and February 2016 (Oliver et al., 2017). A similar event also occurred in this region during austral summer 2017/2018 (Kajtar et al., 2022). MHW conditions persisted for approximately 3 months, with mean Tasman sea surface temperatures 1 to 3 °C above the 1983 to 2012 seasonal climatology (Kajtar et al., 2022). Finally, in austral summer 2021/2022 the position of the EAC jet and its eddies off southeastern Australia had a crucial role in driving MHW/MCS conditions in coastal waters close to Sydney. A MHW occurred in December 2021 and February 2022, while a MCS occurred in January 2022 (Li et al., 2023).

Due to data availability in space and time, many MHW and MCS studies are focused at the surface (e.g. Oliver et al. (2018); Frölicher et al. (2018); Schlegel et al. (2021); Kajtar et al. (2021); Marin et al. (2021)). Schaeffer and Roughan (2017) were amongst the first to analyse MHWs at multiple depths between the surface and the bottom at a 65 and 100 m coastal site off Sydney, Australia. Using in situ data from these sites, they showed that MHWs are often at a maximum intensity below the surface, illustrating the importance of understanding the complex structure of MHWs below the surface.

Some of the longest sub-surface temperature measurements in the southern hemisphere are located at four Australian coastal sites (Roughan et al., 2022a). One site is located on the west coast, influenced by the Leeuwin Current system, and three sites are located on the east coast influenced by the East Australian Current (EAC) system (Roughan et al., 2022a). These temperature records began as early as the 1940s and continue to the present day. The long temperature records at these sites are valuable for investigating ocean temperature variability at many timescales (Hemming et al., 2020; Roughan et al., 2022a, b) over periods of more than 65 years. Additionally they are valuable for the investigation of marine extremes, such as the characteristics of HSs and CSs, and MHWs and MCSs in more recent decades when high-frequency observations became available (Schaeffer and Roughan, 2017).

Here we describe four unique data products; one for each of Australia's long-term national reference stations described in Section 2, that can be used for analysing individual or multiple extreme temperature events, such as HSs, CSs, MHWs and MCSs, at depth levels through the water column. We provide event metrics, such as intensity and duration for example, useful for characterising MHWs and MCSs, and the temperatures and daily climatologies that were used for event detection. Further, we demonstrate the use of these data products by briefly exploring some of the extreme temperature events.

The oceanographic sites are detailed in Section 2, and the steps used to construct the data products are described in Section 3. Example usages of the data products are described in Section 4. MHW and MCS events are explored in Section 5, and a summary is provided in Section 6. User guides on how to load, slice, and cite the data products are provided in the Appendix.

## 2 Oceanographic Sites

We use ocean temperature data at four long-term national reference stations located around Australia (Fig. 1). These sites are unique in Australia and the southern hemisphere in that they have been occupied weekly to monthly since the 1940s/50s (Roughan et al., 2022a; Lynch et al., 2014). The west coast site is close to Rottnest Island (ROT, $32\,°$S) offshore from Perth, Western Australia in approximately 55 m of water and temperatures here are influenced by the Leeuwin current. The southern site is close to Maria Island (MAI, $42.6\,°$S), Tasmania in approximately 90 m of water and is situated in the southern extension region of the EAC. An additional two east coast sites are located close to Sydney (PHA and PHB/PH100, $34.1\,°$S), New South Wales in approximately 50 and 100 m of water, respectively, downstream of the typical EAC separation zone in what are usually eddy dominated waters. The ROT, MAI, PHA and PHB/PH100 sites are the longest sub-surface ocean time-series in Australian coastal waters and are part of Australia's national reference station network; a network of long-term sampling sites (presently 7) around Australia (Lynch et al., 2014).

Unlike most long-term ocean temperature data sets, measurements have been collected at multiple depths through the water column at PHA, PHB/PH100, MAI and ROT since the 1940/50s. The optimal depths (or binned depths) chosen for the data products described here are shown in Fig. 2 and Tab. 1. These optimal depths were determined using binned data to establish the depths where data availability is highest over the full record (Hemming et al., 2020). Originally, sampling was boat-based and weekly to monthly. In 2008/2009, the ROT, MAI, and PHB/PH100 sites were incorporated into the Integrated Marine Observing System (IMOS) National Reference Station network (Lynch et al., 2014), and continuously-recording thermistor moorings have been deployed alongside monthly boat-based sampling. All moored data have continued to be collected through IMOS since 2008/2009 at a sampling interval of 5-60 minutes at multiple depths through the water column. Although we only have bottle and CTD profiles at the PHA site at weekly to monthly sampling intervals, this site has the longest continuous temperature record in Australian waters (1942-present), as well as concurrent biogeochemical sampling. Hence, while we are not typically able to calculate MHWs and MCSs (due to the 5 day duration required by the definition used here) we provide extreme temperature statistics at PHA as it may be useful for investigating the potential impacts of extreme temperatures on biogeochemical processes. A full description of sites PHA, PHB/PH100, MAI, and ROT, their available data, and corresponding metadata is provided by Roughan et al. (2022a).

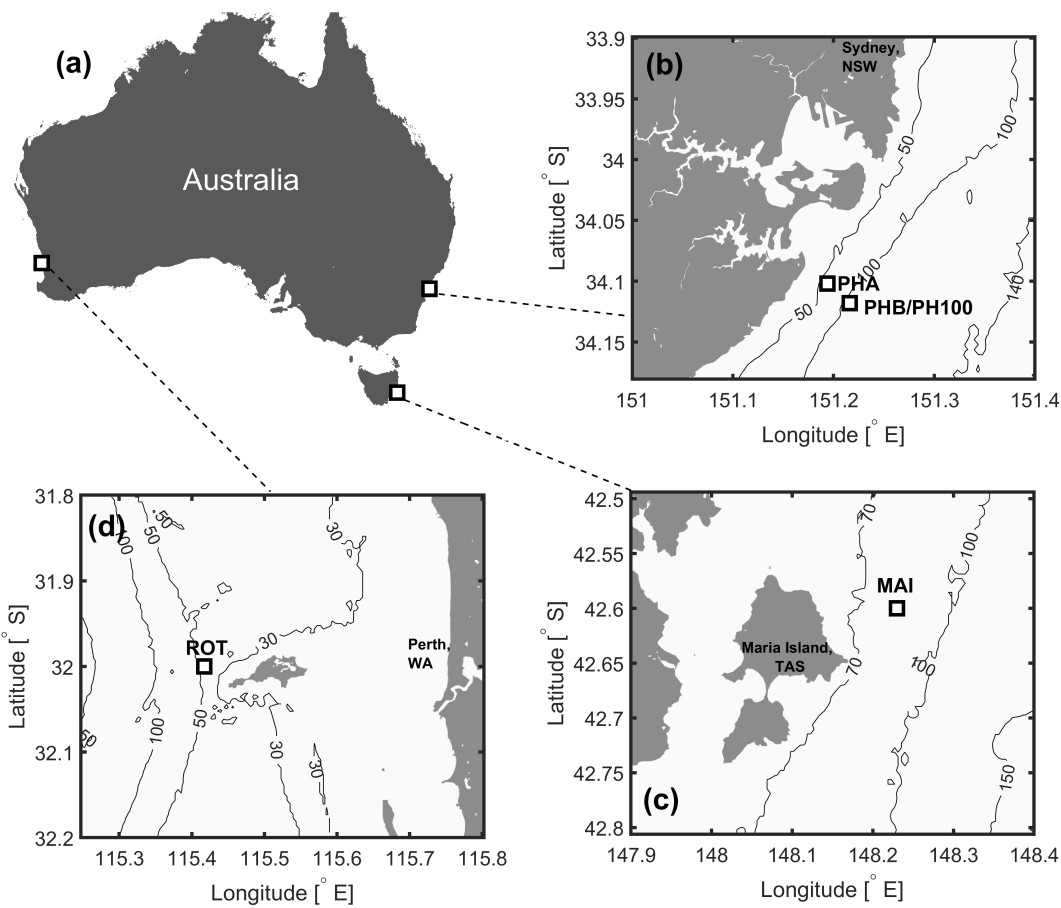

**Figure 1.** a) The 4 long-term oceanographic sampling sites at coastal locations around Australia. b) The Port Hacking stations (close to Sydney) at approximately 50 and 100 m depth (PHA and PHB/PH100). c) The Maria Island (Tasmania) station at approximately 90 m depth (MAI), and d) Rottnest Island (close to Perth, Western Australia) at approximately 55 m depth (ROT). Isobaths (using data from General Bathymetric Chart of the Oceans (GEBCO), Group 2023 (2023)) are also plotted in panels (b), (c) and (d).

## 3 The Data Products

### 3.1 Multi-decadal Time Series Data Products

We build on recently-released aggregated multi-decadal ocean temperature time series products at the four oceanographic sites (Roughan et al., 2022a, b), which we refer to as the Australian Multi-Decadal Ocean Time series (AMDOT) data products. The AMDOT data products at PHA, PHB/PH100, MAI, and ROT include multiple-depth ship-based temperatures from bottle samples and electronic CTD sensor profiles. Several different SeaBird Electronics CTD sensors (e.g. SBE17+, SBE19+, SBE25) have been used to measure temperature and pressure at the sites.

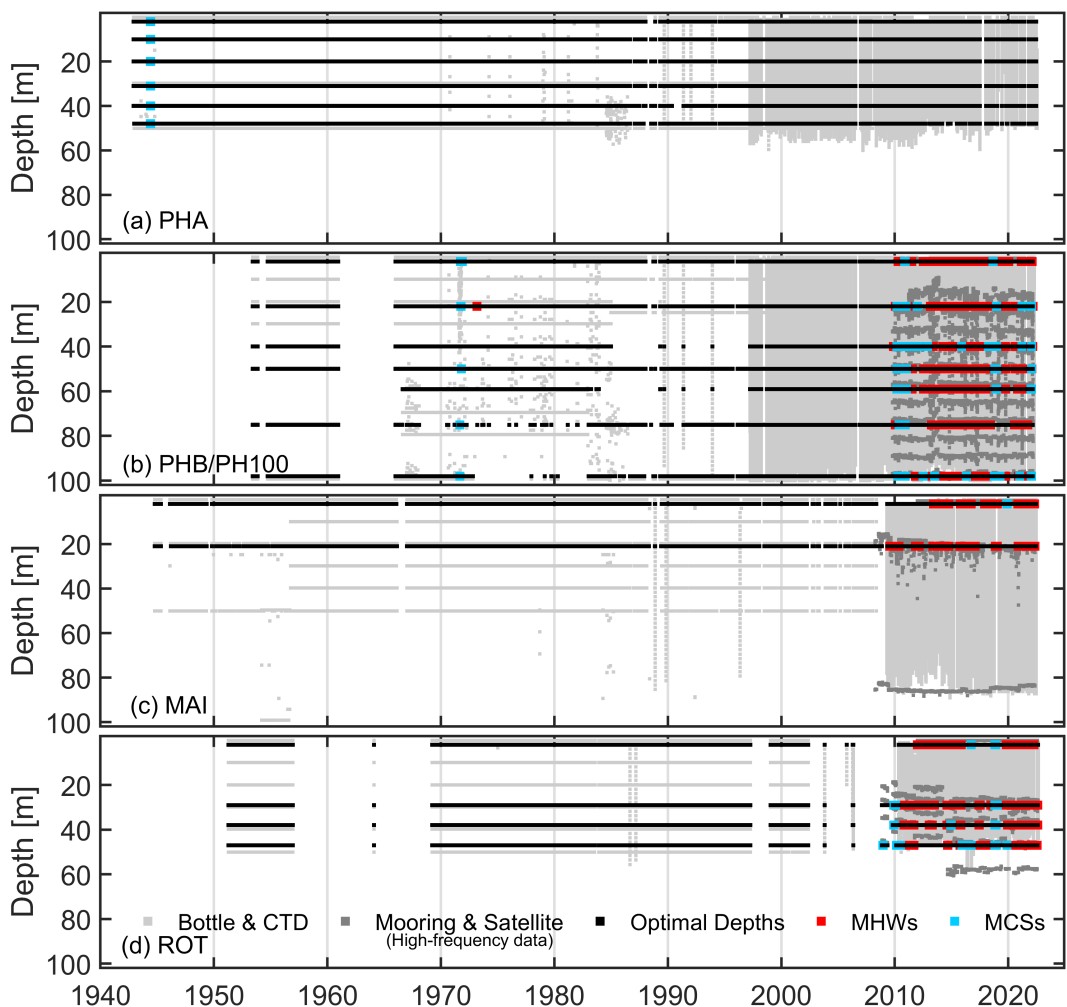

**Figure 2.** Depiction of available temperature data from bottle and CTD profiles (light grey), high-frequency mooring and satellite data (dark grey), and data binned at optimal depths (black) for the a) Port Hacking 50 m (PHA), b) Port Hacking 100 m (PHB/PH100), c) Maria Island (MAI), and d) Rottnest Island (ROT) sites. Data points associated with marine coldspells (MCSs, light blue) and marine heatwaves (MHWs, red) are also shown, and in most cases highlights time periods when high-frequency data exists at the sites.

In addition, the aggregated AMDOT data products at sites PHB/PH100, MAI, and ROT contain multiple-depth moored temperature measurements. At PHB/PH100 Thermistors (Aquatech AQUAloggers 520T or 520PT) were mounted on a mooring line at 8 m intervals through the water column. At approximately 15 to 24 m a Wetlabs water quality meter was deployed between 2010 and 2017, which was later replaced by a SeaBird SBE37 (including a second one approximately 6 m above the bottom). At MAI Wetlabs WQMs and SeaBird SBE37s were mounted on a mooring line at around 20 to 25 m and close to the
bottom (approx. 90 m). At ROT various temperature sensors (e.g. Wetlabs WQMs, Seabird SBE37 and SBE39) were placed on

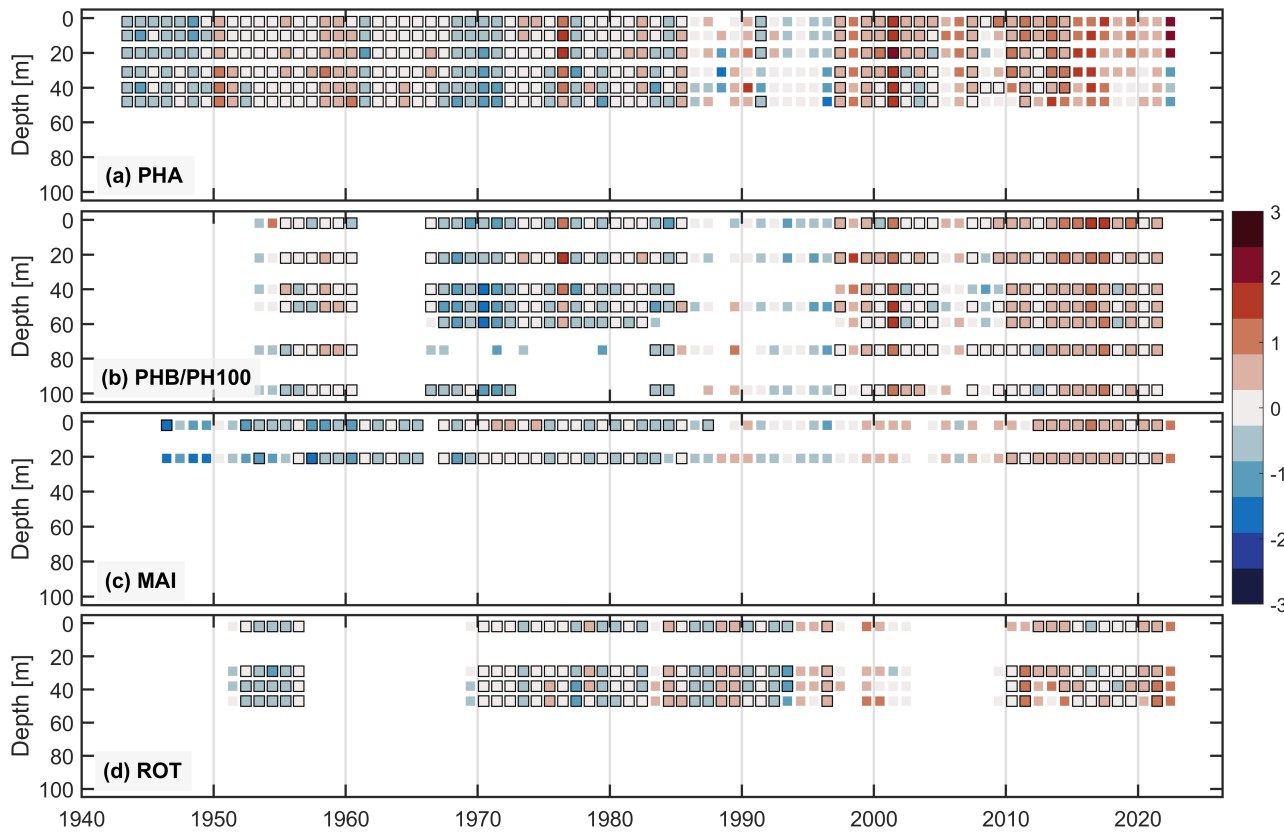

**Figure 3.** Annual median temperature anomalies relative to the mean daily-varying climatology at optimal depths for the a) Port Hacking 50 m (PHA), b) Port Hacking 100 m (PHB/PH100), c) Maria Island (MAI), and d) Rottnest Island (ROT) sites. All years and depths shown have measurements available in at least 4 months of the year, while those years and depths with a solid black outline have measurements available in at least 10 months of the year.

a mooring line at approximately 22, 27, 35, 43, and 55 m depths (varying over time). More information on these configurations can be obtained from Lynch et al. (2014), Chen and Feng (2021), and Roughan et al. (2022a).

At sites PHB/PH100, MAI, and ROT at the surface (0.2 m) after 2012 we include remote sensed sea surface temperature (SST) data from the IMOS Multi-sensor night-time gridded Super-Collated (L3S) SST data composites (Govekar et al., 2022). We use SST at drifting buoy depth (approximately 0.2 m, calculated by adding 0.17 K to skin SST (Govekar et al., 2022)) that has had the sensor specific error statistics (SSES) bias removed from each grid cell. Further, we only use SST associated with 'quality level' flags $\geq 4$ (less likely to be contaminated by clouds). A full description of the data QC, and how these aggregated data products were created is provided by Roughan et al. (2022a), and further information specifically on the SST data and methods is provided by Govekar et al. (2022).

Data availability (Table 1 and Fig 2) is consistent through the water column at PHA because the temperature record consists of bottle and CTD profiles only. However, at PHB/PH100, MAI, and ROT there is increased sub-surface data availability relative to the surface. This is because the satellite-derived SST data that we use starts in 2012 and has some gaps (dependent on clear skies).

The annual median temperatures for each site are shown in Figure 3. The long-term consistency of the temperature data
has been tested through comparisons of measurements overlapping in time and space. At PHB/PH100 and MAI satellite SST agrees well with surface buoy temperature (Hemming, 2023a). At PHB/PH100 there is also good agreement between satellite SST and the nearest sub-surface mooring temperature (Roughan et al., 2022a; Malan et al., 2021), and between water sample and CTD sensor profiles (Hemming, 2023a).

To produce statistics through the water column, we use aggregated temperature data at optimal depth levels for identifying
temperature extremes. These optimal depth levels (listed in Table 1 for each site, and shown in Fig 2) were selected based on the vertical distribution of the data, similar to the method by Roughan et al. (2022a) to produce daily climatologies.

## 3.2 Deriving Extreme Temperature Event Statistics

The method of deriving extreme temperature event statistics is summarised in the flow chart in Fig. 4. We use the open-source Python MHW software package (Oliver, 2020) that uses the widely-adopted definitions (Hobday et al., 2016; Schlegel et al.,
2021) for deriving MHWs and MCSs relative to a baseline climatology. However, instead of producing new daily climatologies using this MHW software package, we adapted the code to use the daily temperature climatologies produced by Roughan et al. (2022a) who use the method described by Hemming et al. (2020) as input for detecting extreme temperature events. These daily temperature climatologies (Roughan et al., 2022b) at the four coastal sites are derived from the aggregated temperatures used here, described in Sect. 3.1. The climatologies were created for each day of the year (01 Jan until 31 Dec, excluding 29 Feb
during leap years) using temperature data within a time-centred moving window of 11 days. To limit warm bias arising from over-representation of mooring data points relative to bottle samples, a bottle-to-mooring ratio was used. The climatologies have been further smoothed using a moving average window of 31 days, following the recommendation of Hobday et al. (2016). As the climatologies are provided at regular 10 m intervals between the shallowest and deepest depth, for the purpose of extreme temperature identification we interpolated the climatology statistics back to optimal depth levels. The reader is
referred to Roughan et al. (2022a) for further details on how these daily climatologies were produced.

We identify HSs as temperatures that are higher than the daily-varying 90[th] percentiles and CSs as temperatures that are lower than the daily-varying 10[th] percentiles. When daily-binned high-frequency fixed time series are available (after 2008 / 2009, see Table 1), we further consider MHWs and MCSs when the spikes last for 5 or more consecutive days (as per the widely adopted definition, where a MHW/MCS event needs to last more than 5 days). As suggested by Hobday et al.
(2016), we ignore intermittent periods of 2 days or less when temperature extremes ease if followed by another MHW or MCS event. These intermittent periods are identified using daily-binned temperatures centred at 12:00 UTC, hence periods of 2 days or less are represented by one less anomalous daily-binned temperature. To increase data coverage, gaps of 2 days or less are filled using linear interpolation prior to detecting MHWs and MCSs using the MHW software package (Oliver, 2020).

**Table 1.** Table showing the number of Marine Heatwaves (MHWs), Marine Cold-Spells (MCSs), Heat Spikes (HSs) and Cold Spikes (CSs) identified at the Port Hacking 50 m and 100 m sites (PHA and PHB/PH100, respectively), Maria Island (MAI), and Rottnest Island (ROT) at multiple depths. HSs and CSs include days when temperatures were higher than the 90th percentile, and lower than the 10th percentile, respectively, and do not include days identified during MHWs and MCSs. The time period that we have high-frequency data and hence when we have most MHWs and MCSs is shown separately to the time period used to additionally detect HSs and CSs. The total record available (excluding interpolated temperatures) at multiple depths as a percentage of the time period between the first and last sampled temperatures is listed for each site. The unique number of days (or daily-binned data points) excluding interpolated temperatures available over the site's record (e.g. between 1951 and 2022 at ROT) are also shown (# Days Sampled).

| Site | Depth | # MHWs | # MCSs | # HSs | # CSs | Record Availability | # Days Sampled |
|---|---|---|---|---|---|---|---|
| PHA | | Time Period 1942 - 2022 | | | | | |
| | 2 | N/A | 1 | 263 | 207 | 7.22% | 2100 |
| | 10 | N/A | 1 | 233 | 224 | 7.17% | 2086 |
| | 20 | N/A | 1 | 256 | 242 | 7.22% | 2101 |
| | 31 | N/A | 1 | 275 | 249 | 7.15% | 2079 |
| | 40 | N/A | 1 | 257 | 260 | 7.06% | 2053 |
| | 48 | N/A | 1 | 234 | 290 | 6.96% | 2025 |
| PHB/PH100 | | Time Period 2009 - 2022 | | | | Time Period 1953 - 2022 | |
| | 2 | 49 | 6 | 426 | 235 | 15.83% | 3984 |
| | 22 | 44 | 9 | 404 | 277 | 18.97% | 4774 |
| | 40 | 45 | 12 | 359 | 308 | 21.42% | 5392 |
| | 50 | 50 | 7 | 358 | 312 | 23.08% | 5810 |
| | 59 | 43 | 6 | 302 | 224 | 18.53% | 4663 |
| | 75 | 40 | 3 | 325 | 214 | 19.56% | 4922 |
| | 98 | 46 | 12 | 281 | 246 | 20.55% | 5172 |
| MAI | | Time Period 2008 - 2022 | | | | Time Period 1944 - 2022 | |
| | 2 | 31 | 1 | 199 | 113 | 10.6% | 3010 |
| | 21 | 30 | N/A | 198 | 136 | 17.53% | 4977 |
| ROT | | Time Period 2008 - 2022 | | | | Time Period 1951 - 2022 | |
| | 2 | 36 | 3 | 288 | 136 | 10.67% | 2783 |
| | 29 | 53 | 7 | 326 | 206 | 19.93% | 5199 |
| | 38 | 39 | 4 | 279 | 208 | 13.93% | 3634 |
| | 47 | 36 | 10 | 209 | 183 | 14.09% | 3675 |

The number of extreme temperature events that we detect depends on the site, depth, and data availability (Table 1, Fig 2). Between 198 and 426 HSs, and between 113 and 312 CSs were detected when considering all sites and depths. When considering individual depths, ROT had the most (53) MHWs at a depth of 29 m and MAI had the least (30) at a depth of 21 m. PHB/PH100 had the most MCSs (12) at depths of 40 and 98 m. Only 1 MCS was detected at MAI at 2 m depth. It was generally not possible to detect MHWs and MCSs at PHA using the Hobday et al. (2016) definition because we only have data that were collected weekly to monthly. However, one MCS was detected at multiple depths in late May / June 1944 when 5 bottle profiles were collected within a period of 8 days.

The AMDOT-EXT data products contain extreme events that sometimes vary over depth. For example, event start and end dates, and their category, can vary as a function of depth (Fig. 5). This is partially because data availability over time is often non-uniform through the water column, but also because the depth structure of MHW events can vary due to ocean dynamics. For example in the coastal Sydney region, Schaeffer et al. (2023) showed that MHWs can be shallow, can extend across the whole water column, or can be sub-surface only. During atmospherically-forced shallow MHWs, for instance, we might expect there to be fewer days when temperatures exceed the $90^{\text{th}}$ percentile at depths below the mixed layer depth when compared with temperatures at the surface (Schaeffer et al., 2023).

## 3.3 Data Products and Variables

We provide one data product at each of the four sites that contains: extreme temperature event information, the temperature data used, and climatology statistics. We refer to these data products as the Australian Multi-Decadal Ocean Time series EXTreme (AMDOT-EXT) data products. Two temperature variables are included in the AMDOT-EXT files; each temperature variable includes the daily-binned temperatures at each depth level ($\pm 3$ m bins), however one temperature variable has gaps of $\leq 2$ days filled. This latter gap-filled temperature variable was used for detecting extreme temperature events. The AMDOT-EXT data products also include the corresponding time and depth (constant over time), and the climatological mean, $10^{\text{th}}$ and $90^{\text{th}}$ percentiles. Table 2 summarises the names of the variables contained in the AMDOT-EXT files. Event information (e.g. event number, duration, intensity etc.) is accessible using matrix variables with dimensions 'TIME' and 'DEPTH'. These matrices have the same dimensions as the temperature data and climatology parameters.

The individual MHW and MCS event numbers and the categories defined by Hobday et al. (2018) and Schlegel et al. (2021) are included in the AMDOT-EXT files, with their variable names listed in Table 2. The MHW/MCS category is presented as flags from one to four representing in order a 'Moderate','Strong','Severe' or 'Extreme' event. Due to the inconsistencies in data sampling over depth and / or regional dynamics, an extreme temperature event may be detected at one depth, but not at another, or the length of an event may vary (Fig 5). Hence, the user should keep in mind that there may be inconsistent event numbers and timings of such events when analysing a particular event over depth.

The AMDOT-EXT data products contain an extreme temperature index variable (*TEMP_EXTREME_INDEX*) that is useful for identifying the type of extreme temperature event: CS (flag 1), MCS (flag 2), HS (flag 11), and MHW (flag 12). For MHWs and MCSs we also provide the mean, maximum, and cumulative intensities, variance of intensity, and event onset and decline rates. The intensity is the temperature anomaly relative to the chosen statistic (e.g. mean climatology) (Hobday et al., 2016).

Over the duration of a MHW/MCS, the cumulative intensity is the sum of the daily temperature intensities, while the variance of intensity is the variation (Hobday et al., 2016). We provide event intensity variables that are relative to both the mean and threshold (e.g. 90[th] percentile) climatology (see Table 2). The onset rate represents the temperature change rate from the start of a MHW/MCS to the maximum intensity (lowest value for MCSs), while the decline rate represents the temperature change rate from the maximum intensity to the end of the MHW/MCS (Hobday et al., 2016). These metrics are calculated using the MHW software package (Oliver, 2020) according to the methodology described by Hobday et al. (2016), detailed in their Table 2.

Some metrics for MHWs and MCSs, such as duration, maximum intensity, and cumulative intensity, for example, are constant over the duration of an event. In the NetCDF files these constant metrics are repeated for each day of an event. For example, a MHW lasting 6 days with a maximum intensity of 23.2 °C will have 6 repeated duration values (e.g. 6,6,6,6,6,6) and 6 repeated maximum intensity values (e.g. 23.2, 23.2, 23.2, 23.2, 23.2, 23.2) representing each date of the MHW.

In addition to the AMDOT-EXT NetCDF data products, we provide CSV files (Hemming, 2023b) containing all of the MHW and MCS event information and characteristics at each of the four sites. Please see Appendix A1 for details on how to cite the AMDOT-EXT data products and the CSV files. Tutorials written in Python, MATLAB, and R demonstrating how to load and use these data products are available here:*OneDriveLink* and are described in Appendices A2 and A3.

### 3.4 Effects of Data Availability and Choice of Baseline

It is important to consider data availability and the daily climatologies that were used when detecting and discussing extreme temperature events. Sub-surface measurements used at PHA, PHB/PH100, MAI, and ROT consist of water samples, electronic Conductivity-Temperature-Depth (CTD) sensor profiles, and at PHB/PH100, MAI, and ROT, mooring data. Data availability over time and in space can be seen in Fig. 2, with notable gaps at PHB/PH100 between the 1960s and 1990s, and at ROT during the 1950s, 1960s and 2000s. Temperature data availability over depth relative to the surface is also shown in Table 1. Additionally, the depths that samples were taken has changed with time, which presents a challenge when it comes to detecting extreme sub-surface temperature events. We partially deal with this by filling gaps of $\leq 2$ days, however, there are many instances of gaps $> 2$ days (Fig 2).

We use daily climatology thresholds incorporating temperature data over the entire record, which is between 70 and 81 years at the four sites. We acknowledge that using a climatology of approximately 30 years is recommended (Hobday et al., 2016; WMO, 2018), and is commonly used for surface MHW studies (e.g. Frölicher et al. (2018); Oliver et al. (2018); Elzahaby and Schaeffer (2019)) where such data exists. Schlegel et al. (2019) showed that using a climatology period $> 30$ years may affect MHW detection in a similar way as using a climatology period $< 30$ years, dependent on environmental multi-decadal variability. Further, Rosselló et al. (2023) highlight the impact of long-term trends on MHWs and argue that a moving baseline is more adequate for detecting consistently rare extremes, although presently there is much debate on this subject (Amaya et al., 2023; Sen Gupta, 2023). We use daily climatologies calculated using the entire temperature record at the long-term sites because 30 year daily sub-surface datasets do not yet exist at our chosen sites. Temperatures at two of the sites: PHB/PH100 and MAI, have increased non-linearly over time, with mean temperatures (dependent on depth) in 2022 approximately 0.5 to

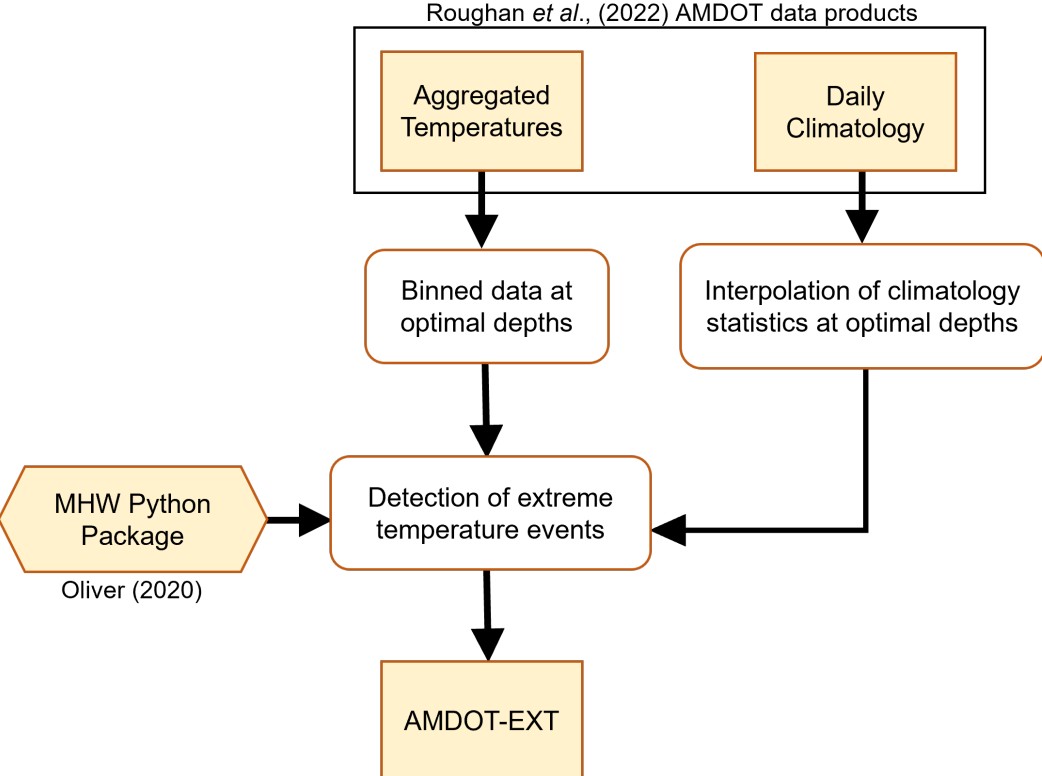

**Figure 4.** A flow chart showing the process of creating the Australian Multi Decadal Ocean Time series EXTreme (AMDOT-EXT) data products.

1.3 °C warmer on average relative to the 1940/50s (Hemming et al., 2023), and annual median temperature anomalies can be seen in Figure 3. Presently, high-frequency sub-surface data at the 4 sites only extend back approximately 13 to 14 years. We acknowledge that the number of extreme temperature events and their metrics will differ in the future when a 30-year daily 215 climatology becomes available, and that it will vary depending on which 30-year period is used (as suggested by Schlegel et al. (2019)).

## 4 Usage

The AMDOT-EXT data products described here can be used to analyse extreme ocean temperature events at the four sites at depths through the water column over time. For example, the *MHW_EVENT_CAT* variable could be used to select 'Strong' 220 MHWs at a particular depth and site. Applying this information to variables *TIME* and *MHW_INTENSITY_MEAN* would select the timings and mean intensities of 'Strong' MHWs. Further, exporting the data during such events might be useful for working with other data sets, for example phytoplankton data for investigating the biological impact of MHWs. Code written in Python,

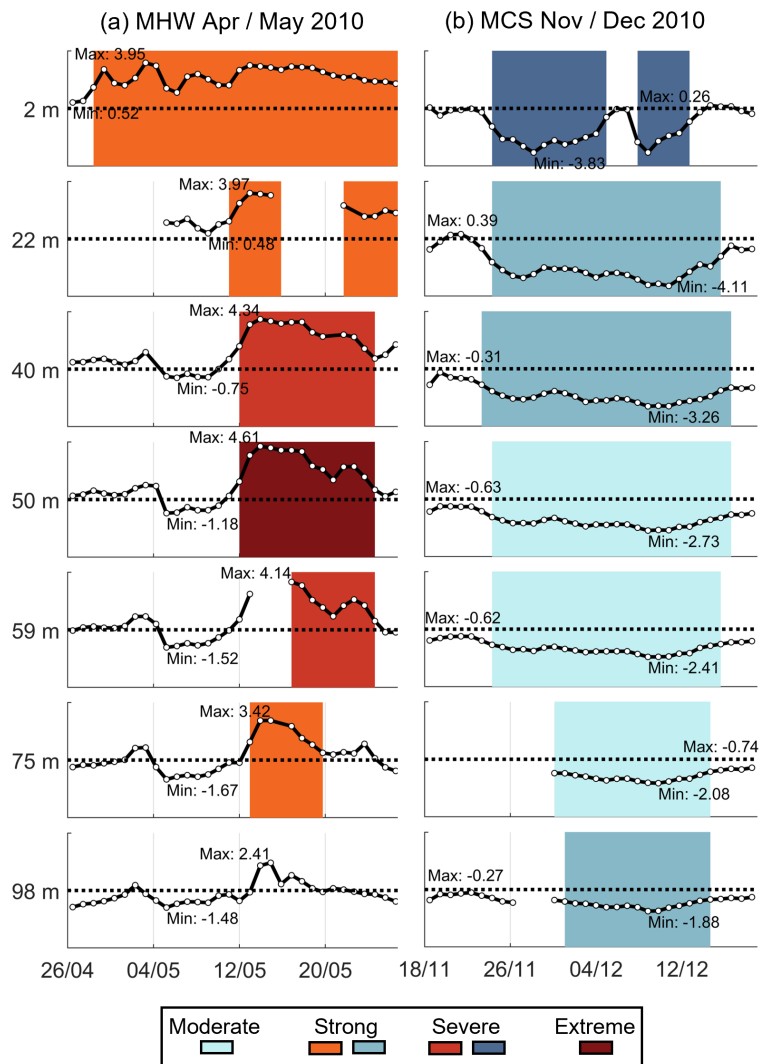

**Figure 5.** Examples of a MHW and MCS over depth at the Port Hacking (PHB/PH100) site: (a) MHW in April / May 2010, and (b) MCS in November / December 2010. The daily-varying temperature anomalies relative to the mean (black line with white dots) are shown for each depth. Time periods when MHWs or MCSs occur are shaded, with colors indicating the category ('moderate', 'strong', 'severe', and 'extreme') as shown in the legend. The maximum and minimum temperature anomaly relative to the mean for each time period and depth is also shown.

MATLAB, and R demonstrating how to load and slice the AMDOT-EXT data products is available here:*OneDriveLink*, and further details are provided in Appendices A2 and A3.

## 5 Marine Heatwaves and Cold-spells

As shown in Table 1, MHWs and MCSs were most often detected at sites and depths that had high-frequency temperature data (typically between 2008/2009 and 2022). In Figure 6 we explore the most intense events (chosen based on cumulative intensity) between 2008/2009 and 2022 at the long-term sites: PHB/PH100, MAI, and ROT. The most intense MHWs were at approximately 20 m depth at sites PHB/PH100 and MAI in 2015 and 2015/16, respectively (Fig 6a,b). These coastal events coincided with the far-reaching MHW conditions in the Tasman Sea related to the EAC (Oliver et al., 2017). At ROT the most intense MHW between 2008 and 2022 was at 29 m in 2011 (Fig 6c) at the time of the well-documented 'Ningaloo' La Niña MHW (Feng et al., 2013; Wernberg et al., 2013; Pearce and Feng, 2013). On average the intensity of the 2015 MHW at PHB/PH100 was 2.4 °C above the mean climatology and lasted 50 days, while the MAI and ROT MHWs were on average 2.2 and 1.9 °C above the mean climatology, and lasted 151 and 60 days, respectively.

The most intense MCSs between 2008/2009 and 2022 were in 2010 and 2015 at PHB/PH100 and ROT, respectively (Fig 6d,f). At PHB/PH100 and ROT, these events were 3.1 and 1.8 °C cooler than the mean climatology on average, and lasted 22 and 25 days, respectively. At MAI we detected 1 MCS between 2008 and 2022 lasting 5 days in late 2019 which was 1.6 °C cooler than the mean climatology on average.

Information relating to the longest MHWs between 2008/2009 and 2022 at sites PHB/PH100, MAI, and ROT is provided in Table 3. The longest MHW detected lasted 151 days at MAI (same as in Fig 6b) between 15/12/2015 and 13/05/2016, while the longest MHWs at PHB/PH100 and ROT lasted 54 and 60 days, respectively. The longest ROT MHW was also the most intense, as described above and shown in Fig 6c. The cumulative intensities of these MHWs were between 92.4 and 330 °C days. The longest MCSs between 2008/2009 and 2022 lasted 24 and 25 days in 2010 and 2015/2016 at PHB/PH100 and ROT, respectively, and just 5 days in 2019 for a single MCS at MAI as described above. The cumulative intensities at PHB/PH100 and ROT for these MCSs were -61 and -44.1 °C days, respectively. The MAI MCS had a cumulative intensity of -7.8 °C days. Almost all of the longest MHWs and MCSs were classified as a 'Strong' event, and occurred sub-surface.

## 6 Summary

Extreme temperature events have been identified and characterised at multiple depths between the surface and the bottom at four oceanographic sites: PHA, PHB/PH100, MAI, and ROT using long-term ocean temperature records. We provide four data products - one data product per oceanographic site - including indices for identifying extreme temperature events, their characteristics, and corresponding temperature time series and daily climatology statistics. We refer to these new data products as the Australian Multi-Decadal Ocean Time series EXTreme (AMDOT-EXT) data products.

These AMDOT-EXT data products are freely accessible at the Australia Ocean Data Network thredds server, and we provide code tutorials in Appendix A demonstrating how to load the data products, use the NetCDF variables, and export the data as CSV files *citation to be finalised after the review phase*. We also provide CSV files containing all of the MHW and MCS event information and characteristics at each of the four sites. Refer to the code and data availability section below for information on how to access the NetCDF and CSV files.

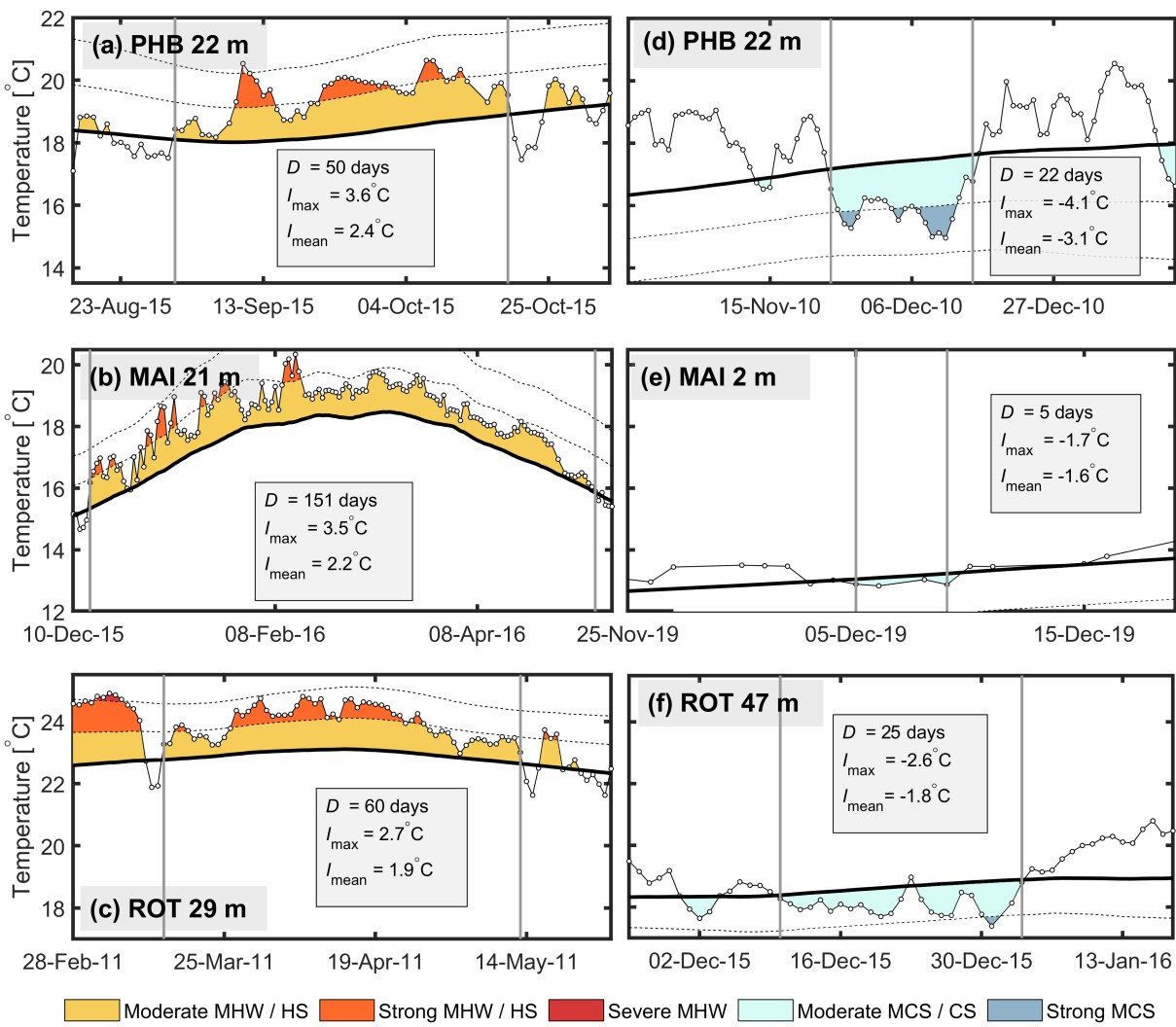

**Figure 6.** Examples of a)-c) Marine Heatwaves / Heat Spikes (MHWs/HS) and d)-f) Marine Cold-spells / Cold Spikes (MCSs / CSs,) with the maximum cumulative intensity at the long-term sites PHB/PH100, MAI, and ROT. The area underneath the portion of the time series experiencing an extreme event is filled according to the category: 'Moderate MHW / HS' (yellow), 'Strong MHW / HS' (orange), 'Severe MHW' (red), 'Moderate MCS / CS' (light blue), and 'Strong MCS' (dark blue). The metrics shown are duration ('D'), maximum intensity ('$I_{\mathrm{max}}$'), and mean intensity ('$I_{\mathrm{mean}}$'). The time period of each MHW or MCS is indicated by vertical grey lines. Each panel covers a different time period, hence the x-limits are not the same in each panel.

Using the AMDOT-EXT data products, we compare the number of extreme temperature events at the four sites, and explore the most intense and longest MHWs and MCSs, which were often below the surface. These data products can be used for characterising and assessing the impacts of MHWs and MCSs in coastal waters around Australia.

The AMDOT-EXT data products highlight the challenges faced when detecting extreme temperature events using sporadic long time series. Hence, it is vital that high temporal resolution measurements at multiple depths (i.e. moored thermistor data) continue into the future to fully understand extreme temperature events and their variability in the face of rapid environmental change.

*Code and data availability.* The AMDOT-EXT data products are available as NetCDF files (https://doi.org/10.26198/wbc7-8h24) (Hemming, 2023b), and variables contained in the NetCDF files are listed in Table 2. Any and all use of the AMDOT-EXT data products provided here must include a citation (See Appendix A1). Any updated data products, and or potential new products (e.g. at other sites or using other ocean variables) will be hosted at the same location. Therefore, it is advised that data users seek the latest data product version.

We provide basic scripts in MATLAB, Python and R demonstrating how to download and load the data products, use the NetCDF variables,
produce plots, and export the data as CSV files. These scripts are available online at Zenodo here:*OneDriveLink* and are available to use under a Creative Commons Attribution 4.0 International license (CC BY 4.0). Please see Appendix A below for more details.

## Appendix A: User Guide and Tutorials

### A1    Citing AMDOT-EXT Data Products

Any and all use of the AMDOT-EXT data products or accompanying event summary CSV files described here must include:

– a citation to this paper,

   – a reference to the data citation as written in the NetCDF file attributes and as follows: Hemming, MP. et al. (2023) "Multi-decadal time series of sub-surface marine heatwaves and cold-spells in Australian shelf waters", Australian Ocean Data Network, https://doi.org/10.26198/wbc7-8h24.

   – the following acknowledgement statement: Data were sourced from Australia's Integrated Marine Observing System
(IMOS) - IMOS is enabled by the National Collaborative Research Infrastructure Strategy (NCRIS).

### A2    Loading AMDOT-EXT Data Products and Saving as CSV Files

The tutorial script 'get_DataProducts' available on Zenodo (*OneDriveLink*) demonstrates how to do the following:

   – Load the MAI AMDOT-EXT data product using OPeNDAP,

   – Select data during a MHW event,

– export the data as a Comma Separated Values (CSV) file

A version of this tutorial is available for use with R, Python, and MATLAB, and can be modified to use any AMDOT-EXT data product.

### A3 Slicing AMDOT-EXT Data Product Variables Based on Event Characteristics

The tutorial script 'slice_DataProducts' available on Zenodo (*OneDriveLink*) demonstrates how to do the following:

– Load the PH100 AMDOT-EXT data product using OPeNDAP,

   – select data during 'strong' category MHWs at a depth of 22 m,

   – calculate and display mean characteristics during 'strong' category MHWs at this depth,

   – save the sliced data set as either a NetCDF, MAT (MATLAB) or rdata (R) file, as well as a Comma Separated Values (CSV) file

A version of this tutorial is available for use with R, Python, and MATLAB, and can be modified to slice any AMDOT-EXT data product based on any event characteristic.

*Author contributions.* MH and MR conceived the study. MH developed the data products and analysed the data, wrote the code tutorials, created the figures, and drafted the manuscript. MR led the project and obtained the funding. AS tested and reviewed the NetCDF files. MR and AS reviewed the manuscript.

*Competing interests.* We declare that no competing interests are present.

*Acknowledgements.* We acknowledge the foresight of CSIRO Marine Research for instigating the data collection in the 1940s and its continuation in recent decades through IMOS. We are indebted to everyone involved in the data collection, including the Australian National Mooring Network and National Reference Station field teams, and former CSIRO technical staff for the on-going mooring deployments, boat-based hydrographic sampling and data processing since the 1940s. We acknowledge Tim Moltmann, former IMOS director, for his 305 unwavering belief that data should be open and accessible - which prompted this work. We thank two anonymous reviewers for their time and helpful comments, and a final thanks to Natalia Atkins and Laurent Besnard at the Australian Ocean Data Network for creating the data set DOI and reviewing the NetCDF files. Data were sourced from Australia's Integrated Marine Observing System (IMOS) – IMOS is enabled by the National Collaborative Research Infrastructure Strategy (NCRIS).

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

**Table 2.** A table summarising the variables and their dimensions contained in each AMDOT-EXT NetCDF file with corresponding description.

| Variable | Dimensions | Description |
|---|---|---|
| TIME | TIME | An array containing time information (days since 1950-01-01). |
| DEPTH | DEPTH | An array containing the chosen depth levels for deriving event statistics. |
| TEMP | TIME, DEPTH | A matrix containing the binned temperatures over the entire record for each depth level (Roughan et al., 2022a) |
| TEMP_INTERP | TIME, DEPTH | A matrix containing the same binned temperatures as in variable 'TEMP' but with gaps of $\leq 2$ days filled. These temperatures are used to derive event statistics. |
| TEMP_<event type>_SPIKE<br><br>(e.g. TEMP_HEAT_SPIKE) | TIME, DEPTH | A matrix containing all temperatures that exceed the percentile limit ($90^{\text{th}}$ percentile for heat spikes, and $10^{\text{th}}$ percentile for cold spikes). This includes temperatures during MHWs / MCSs too. |
| TEMP_EXTREME_INDEX | TIME, DEPTH | An extreme temperature event index useful for selecting specific event types. Numbers 1, 2, 11 and 12 correspond with CSs, MCSs, HSs and MHWs, respectively. |
| TEMP_MEAN | TIME, DEPTH | The daily mean climatology produced by Roughan et al. (2022a), and used for deriving event statistics here, for each date contained in TIME. |
| TEMP_PER10 | TIME, DEPTH | The daily $10^{\text{th}}$ percentiles produced by Roughan et al. (2022a), and used for deriving event statistics here, for each date contained in TIME. |
| TEMP_PER50 | TIME, DEPTH | The daily $50^{\text{th}}$ percentiles produced by Roughan et al. (2022a), and used for deriving event statistics here, for each date contained in TIME. |
| TEMP_PER90 | TIME, DEPTH | The daily $90^{\text{th}}$ percentiles produced by Roughan et al. (2022a), and used for deriving event statistics here, for each date contained in TIME. |
| <event type>_EVENT_NUMBER<br><br>(e.g. MHW_EVENT_NUMBER) | TIME, DEPTH | A matrix containing the event numbers of a chosen event type (e.g. MHW or MCS) for each depth level. |
| <event type>_EVENT_CAT<br><br>(e.g. MHW_EVENT_CAT) | TIME, DEPTH | A matrix containing the maximum event category number i.e 1 to 4: 'Moderate', 'Strong', 'Severe' and 'Extreme' for each depth level. |
| <event type>_EVENT_ONSET_RATE<br><br>(e.g. MHW_EVENT_ONSET_RATE) | TIME, DEPTH | A matrix containing the onset rate of a chosen event type (e.g. MHW or MCS) for each depth level. |
| <event type>_EVENT_DECLINE_RATE<br><br>(e.g. MHW_EVENT_DECLINE_RATE) | TIME, DEPTH | A matrix containing the decline rate of a chosen event type (e.g. MHW or MCS) for each depth level. |
| <event type>_EVENT_INTENSITY_<intensity type><br><br>(e.g. MHW_EVENT_INTENSITY_MEAN) | TIME, DEPTH | A matrix containing either the mean, max, cumulative, or variance of intensity for a chosen event type (e.g. MHW or MCS) for each depth level. Intensity is calculated relative to the mean daily climatology (Roughan et al., 2022a). |
| <event type>_EVENT_INTENSITY_<intensity type>_RELPERC | TIME, DEPTH | A matrix containing either the mean, max, cumulative, or variance of intensity for a chosen event type (e.g. MHW or MCS) for each depth level. Intensity is calculated relative to the daily $10^{\text{th}}$ or $90^{\text{th}}$ percentiles (Roughan et al., 2022a) depending on |

**Table 3.** The date range, duration, depth, maximum classification, and cumulative intensity ($I_c$) for the longest marine heatwaves (MHWs) and marine cold-spells (MCSs) at sites PHB/PH100, MAI, and ROT.

| Site | Longest MHWs Date Range | Duration days | Depth m | Max. Classification | $I_c$ °C days | Longest MCSs Date Range | Duration days | Depth m | Max. Classification | $I_c$ °C days |
|---|---|---|---|---|---|---|---|---|---|---|
| PHB /PH100 | 19-Jun to 11-Aug-2017 | 54 | 59 | Strong | 92.4 | 23-Nov to 16-Dec-2010 | 24 | 40 | Strong | -61 |
| MAI | 15-Dec-2015 to 13-May-2016 | 151 | 21 | Strong | 330 | 5-Dec to 9-Dec-2019 | 5 | 2 | Moderate | -7.8 |
| ROT | 15-Mar to 13-May-2011 | 60 | 29 | Strong | 116 | 10-Dec-2015 to 03-Jan-2016 | 25 | 47 | Strong | -44.1 |