# Peer review of "Exploring Multi-decadal Time Series of Temperature Extremes in Australian Coastal Waters"

_Earth System Science Data, 2023_

## Author Comment (AC1)

The paper by Hemming and coauthors describes an application of the multi-decadal in-situ observational dataset of marine temperatures (AMDOT) for the detection of extremes (AMDOT-EXT).

We thank the reviewer for their insightful comments.

While I appreciate the importance of extended observational records for assessing long-term changes, I am not sure to understand how to interpret MHW/MCS numbers with less-than-daily frequency data. From the images included, it seems that high-impact events can be detected only over the last ~20 years (compare Table 3 and Figure 4), even if at l100 an event in 1944 is described.

It is true that MHW and MCS events can only be identified as per the Hobday (daily) definition when daily-binned data are available as they define MHW/MCSs as discrete prolonged events lasting 5 days or more. Hence, we can mostly derive MHW/MCS characteristics when we have daily mooring and satellite measurements since 2008/2009. In the absence of daily data, we can also consider temperature 'spikes', which are also interesting, especially considering the long-term time-series.

We included text stating this in the submitted manuscript. For example, in Section 3.2 (now edited):

"We identify HSs as temperatures that are higher than the daily-varying 90$^{th}$ percentiles and CSs as temperatures that are lower than the daily-varying 10$^{th}$ percentiles. When continuous daily fixed time series are available (after 2008 / 2009, see Table 1), we further consider MHWs and MCSs when the spikes last for 5 or more consecutive days (as per the widely adopted definition, where a MHW/MCS event needs to last more than 5 days)."

Please also refer to our response to your related comment below.

I find Table 1 and the duration of the MHW/MCS records not really clear. Please explain (with a table or figure) the periods for which high-frequency extreme statistics are available.

We incorrectly used the HS/CS time periods for MHW/MCSs and vice versa (see also response to R2). We have corrected the time periods in the updated Table 1 which will improve clarity.

We have changed the colour of the data points shown in Figure 2 (also copied below) that were used to derive MHW/MCS events, and we have added a corresponding label to the legend. These coloured data points highlight when high-frequency extreme statistics are available. Additionally, we have added '(high-frequency data)' underneath 'Mooring & Satellite' in the legend.

[Figure]

**Figure 2.** Depiction of available temperature data from bottle and CTD profiles (light grey), mooring and satellite (dark grey), and data binned at optimal depths (black) for a) Port Hacking 50 m (PHA), b) Port Hacking 100 m (PHB/PH100), c) Maria Island (MAI), and d) Rottnest Island (ROT). Data points associated with marine coldspells (MCSs, light blue) and marine heatwaves (MHWs, red) are also shown, and in most cases highlights the time periods when continuous data exists at the sites.

Please note that we have updated the AMDOT-EXT data products. Refer to our response to R2 for more details.

Moreover, some information on the changes in climatology should be provided, since using the whole record likely lead to more events in recent years.

This is indeed an area of ongoing research, and there is much debate in the community on which way is best (e.g. Amaya et al., 2023; Sen Gupta, 2023; Rosselló et al., 2023), and we believe the answer is dependent on the question being asked. In our case we have chosen to use the full record and have been very clear about the time period used, as stated in a dedicated section 3.4 'Effects of Data Availability and Choice of Baseline'. We do this as it would be challenging to compare events using a 30-year climatology below the surface due to a lack of data at our sites.

We have edited Section 3.4 for additional clarity:

*"We acknowledge that using a climatology of approximately 30 years is recommended (Hobday et al.,2016; WMO, 2018), and is commonly used for surface MHW studies (e.g. Frölicher et al. (2018); Oliver et al. (2018); Elzahaby and Schaeffer (2019)) where such data exists. Schlegel et al. (2019) showed that using a climatology period > 30 years may affect MHW detection in a similar way as using a climatology period < 30 years, dependent on environmental multi-decadal variability. Further, Rosello et al. (2023) highlight the impact of long-term trends on MHWs and argue that a moving*

*baseline is more adequate for detecting consistently rare extremes, although presently there is much debate on this subject (Amaya et al. (2023); Sen Gupta (2023)). We use daily climatologies calculated using the entire temperature record at the long-term sites because 30 year daily sub-surface data sets do not yet exist. At this time, high-frequency sub-surface data at the 4 sites only extend back approximately 13 to 14 years. We acknowledge that the number of extreme temperature events and their metrics will differ in the future when a 30-year daily climatology becomes available, and that it will vary depending on which 30-year period is used (as suggested by Schlegel et al 2019)."*

Amaya, Dillon J., Michael G. Jacox, Melanie R. Fewings, Vincent S. Saba, Malte F. Stuecker, Ryan R. Rykaczewski, Andrew C. Ross, et al. "Marine Heatwaves Need Clear Definitions so Coastal Communities Can Adapt." Nature 616, no. 7955 (April 2023): 29–32. https://doi.org/10.1038/d41586-023-00924-2.

Sen Gupta, Alex. "Marine Heatwaves: Definition Duel Heats Up." Nature 617, no. 7961 (May 16, 2023): 465–465. https://doi.org/10.1038/d41586-023-01619-4.

Rosselló, P., Pascual, A., & Combes, V. (2023). "Assessing marine heat waves in the Mediterranean Sea: a comparison of fixed and moving baseline methods," Frontiers in Marine Science, Volume 10, Page 1168368

Analysis quantifying the consequences of selecting different climatology periods is out of scope for this paper as our focus is to describe and explore the new AMDOT-EXT data products. This additional analysis would warrant a whole new paper, when considering work undertaken by e.g. Rosello *et al.*, (2023) and Schlegel *et al.*, (2019). These studies suggest that time series of different lengths will have different trends, and hence the climatologies will differ.

We have hence not included additional text on this in the manuscript.

Background on other possible data sources in the area should be given. For example, no long-term measurement is available for Northern Australia? Data for the last ~20 years are likely available for more sites.

The long-term sites along the southeastern coast of Australia are the focus of this paper. While there are some long-term data sites in northern Australia, we are not the custodians of it. However, to provide some additional context we have added the following sentence in Section 2:

*"The ROT, MAI, and PHB/PH100 sites are the longest sub-surface ocean time-series in Australian coastal waters and are part of Australia's national reference station network; a network of long-term sampling sites (presently 7) around Australia (Lynch et al., 2014)."*

The authors rely on Roughan 2022b for too many details. More information on the observational methods need to be provided. For example, no discussion on the long-term consistency of these records is provided.

We have provided more information on the observational methods in Section 3.1:

*"Several different SeaBird Electronics CTD sensors (e.g. SBE17+, SBE19+, SBE25) have been used to measure temperature and pressure at the sites."*

*"At PHB/PH100 Thermistors (Aquatech AQUAloggers 520 T or 520PT) were mounted on a mooring line at 8 m intervals through the water column, while at MAI Wetlabs WQMs and SeaBird SBE37s were mounted on a mooring line at around 20 to 25 m and close to the bottom (approx. 90 m). At ROT various temperature sensors (e.g. Wetlabs WQMs, Seabird SBE37 and SBE39) were placed on a mooring line at approximately 22, 27, 35, 43, and 55 m (varying over time)."*

*"We use SST at drifting buoy depth (approximately 0.2 m) that has had the sensor specific error statistics (SSES) bias removed from each grid cell. Further, we only use SST associated with `quality level' flags >= 4 (less likely to be contaminated by clouds). A full description of the data QC, and how these aggregated data products were created is provided by Roughan et al. (2022c), and further information specifically on the SST data and methods is provided by Govekar et al. (2022)."*

*"The long-term consistency of the temperature data has been tested through comparisons of measurements overlapping in time and space. At PHB/PH100 and MAI satellite SST agrees well with surface buoy temperature (Hemming, 2023a). At PHB/PH100 there is also good agreement between satellite SST and the nearest subsurface mooring temperature (Roughan et al., 2022a; Malan et al., 2021), and between water sample and CTD sensor profiles (Hemming, 2023a)."*

Familiarity with the Hobday's terminology is also assumed, please explain what technical terms (e.g., cumulative intensity) mean. Please carefully check the units.

We have added the following text to Section 3.3:

*"The intensity is the temperature anomaly relative to the chosen statistic (e.g. mean climatology) (Hobday et al., 2016). Over the duration of a MHW/MCS, the cumulative intensity is the sum of the daily temperature intensities, while the variance of intensity is the variation (Hobday et al., 2016). We provide event intensity variables that are relative to both the mean and threshold (e.g. 90th percentile) climatology (see Table 2). The onset rate represents the temperature change rate from the start of a MHW/MCS to the maximum intensity, while the decline rate represents the temperature change rate from the maximum intensity to the end of the MHW/MCS (Hobday et al., 2016)."*

We have also modified the units for cumulative intensity to be '°C days' throughout the manuscript.

The authors mention 'optimal depths', but it is unclear to me how these are defined.

We have added the following text to Section 2:

*"These optimal depths were determined using binned data to establish the depths where data availability is highest over the full record (Hemming et al., 2020)."*

The depth-resolved information seems an asset, please include some illustration of an event detected at multiple depths.

We have added a new figure showing a marine heatwave and a marine cold spell over depth at the PHB/PH100 site, which will be included in the updated manuscript, and which is also copied below.

[Figure]

**Figure 4.** Examples of a MHW and MCS over depth at the Port Hacking (PHB/PH100) site: (a) MHW in April / May 2010, and (b) MCS in November / December 2010. The daily-varying temperature anomalies relative to the mean (black line with white dots) are shown for each depth. Time periods when MHWs or MCSs occur are shaded, with colors indicating the category ('moderate', 'strong', 'severe', and 'extreme') as shown in the legend. The maximum and minimum temperature anomaly relative to the mean for each time period and depth is also shown.

'Code and data availability' and A1 contents look too similar. Revise.

We have modified the first paragraph in 'Code and data availability' as follows:

*"The AMDOT-EXT data products are available as NetCDF files (https://doi.org/10.26198/wbc7-8h24) (Hemming, 2023), and variables contained in the NetCDF files are listed in Table 2. Any and all use of the AMDOT-EXT data products provided here must include a citation .  Any updated data products, and or potential new products (e.g. at other sites or using other ocean variables) will be hosted at the same location. Therefore, it is advised that data users seek the latest data product version."*

And we have removed the following from Appendix A1:

*""*

The provided codes can be useful to know how to open the data but seems quite basic.

Thank you for looking at the code. In addition to opening the data sets they also demonstrate how to extract required information, save output in different formats (e.g. CSV), and how to slice and analyse the event characteristics (See Appendix A3 for a summary of this code). It is out of scope for this data paper to provide more complex code tutorials.

I do not understand how you deal with event statistics: categories can be defined at daily frequency, but statistics are for individual events (e.g., cumulative intensity). How do you save them as a function of TIME and DEPTH? The same values are repeated for all the days during a MHW/MCS?

We save event-constant statistics (e.g. cumulative intensity) for a MHW/MCS as the reviewer suggested. The same values are repeated for all the days of the event.

We have added the following text clarifying this in Section 3.3:

"*Some metrics for MHWs and MCSs, such as duration, maximum intensity, and cumulative intensity, for example, are constant over the duration of an event. In the NetCDF files these constant metrics are repeated for each day of an event. For example, a MHW lasting 6 days with a maximum intensity of 23.2 °C will have 6 repeated duration values (e.g. 6,6,6,6,6,6) and 6 repeated maximum intensity values (e.g. 23.2, 23.2, 23.2, 23.2, 23.2, 23.2) representing each date of the MHW.*"

l27 please give details on the depth they analysed

l30 the last part of the sentence is a repetition

In response to the two comments above, we have edited this sentence as follows:

"*… were amongst the first to analyse MHWs  at multiple depths between the surface and the bottom at a 65 and 100 m coasta  site off Sydney, Australia.  Using in situ data from these sites  …*"

l37 if from the early 40s, why not ~80 years?

It is true that we have data from the early 1940s at PHA and MAI, but at ROT and PHB we have data from the 1950s. Hence, to include all sites we stated 'more than 65 years'.

We have now modified this sentence to include:

"*… over  70 to 81 year periods.*"

Fig 1 please explain what isolines are (bathymetry? Derived from?)

We have added the following to the caption for Fig 1:

"* … Isobaths (data from General Bathymetric Chart of the Oceans (GEBCO), Weatherall et al. (2023)) are also plotted in panels (b), (c) and (d).*"

Table 1 for 2m depth, I would show not the relative (i.e., always 100%) but the absolute available fraction (maybe on monthly or weekly basis). This would help interpreting other relative fractions.

We have changed the 2m depth relative % to the absolute available fraction as suggested. In other words, the total record available at 2 m depth as a percentage of the time period between the first and last daily-binned temperature.

We have also changed the relative % record available text as follows: e.g. '99%' -> '- 1%', '129.73%' -> '+ 29.73%', which we think is clearer.  These percentages represent sub-surface data availability relative to 2 m depth. The column heading has also been changed from 'Relative % Record Available' to 'Record availability' and further text clarifying this column has been added to the table caption.

Moreover, I do not like the fact that the table is an image - please check the editorial guidelines.

The table is an image during the review period only as it is easier for us to modify it. We will include a LaTeX table in the final version of the manuscript.

l55 please provide information on the instrumentation, I guess it varied over time leading to varying uncertainty. How these levels are related to local conditions such as the mixed layer depth?

It is beyond the scope of this manuscript to explore how the temperature varies in response to local conditions such as MLD.  At the reviewers request we have provided more information on the instrumentation in Section 3.1:

" Several different *SeaBird Electronics CTD sensors (e.g. SBE17+, SBE19+, SBE25) have been used to measure temperature and pressure at the sites.*"

"*At PHB/PH100 Thermistors (Aquatech AQUAloggers 520 T or 520PT) were mounted on a mooring line at 8 m intervals through the water column, while at MAI Wetlabs WQMs and SeaBird SBE37s were mounted on a mooring line at around 20 to 25 m and close to the bottom (approx. 90 m).  At ROT various temperature sensors (e.g. Wetlabs WQMs, Seabird SBE37 and SBE39) were placed on a mooring line at approximately 22, 27, 35, 43, and 55 m depths (varying over time)." More information on these configurations can be obtained from Lynch et al. (2014); Chen and Feng (2021); Roughan et al. (2022c).*"

"*The long-term consistency of the temperature data has been tested through comparisons of measurements overlapping in time and space. At PHB/PH100 and MAI satellite SST agrees well with surface buoy temperature (Hemming, 2023a). At PHB/PH100 there is also good agreement between satellite SST and the nearest subsurface mooring temperature (Roughan et al., 2022a; Malan et al., 2021), and between water sample and CTD sensor profiles (Hemming, 2023a).*"

l66 define CTD

We now define 'CTD' on this line.

l68 how L3S differs from L3? Define 'QC'

'L3' is used to identify gridded SST data, while the 'S' in 'L3S' represents gridded Super-Collated SST composites as per Govekar et al., (2022). We have added the following to this line:

" *… (SST) data from the IMOS Multi-sensor night-time gridded Super-Collated (L3S) SST data composites (Govekar et al., 2022)*"

l75 'free' => 'open source'

We have changed this.

l104 how do you identify the MLD?

We do not identify the MLD in this work as that is beyond the scope of this study. Instead on this line we are referring to the MLD in an example to demonstrate that event variability can be real, as well as due to data availability. We have therefore not added additional text to the submitted original manuscript related to this comment.

l105 missing citation

We have now added this citation to the manuscript.

l123 and elsewhere (e.g. Table 2) I would use italics or a typewriter font for variable names, to distinguish them from MHW, MCS,... - check journal guidelines

We agree that italics might be better for variable names, as the journal guidelines suggest using italics for words, phrases, and abbreviations that cannot be found in the English dictionary. We have therefore changed all variable names to italics in the manuscript.

l130 Excel is not really needed for csv files, they are simply text files (as they are not spreadsheets). Moreover, Excel is not open nor free - suggest providing open-source alternatives.

We have replaced 'spreadsheets' with 'files' throughout the manuscript. The reviewer is correct that CSV files can be opened by many different software. We have removed reference to any single software in the updated manuscript.

l132, 157, 221 what is 'onedrivelink'? Why not using the website you cite?

We agreed with Copernicus ESSD upon submission that for the review process we would supply the tutorials via a OneDrive link for the reviewers to access only. We will upload the tutorials to Zenodo with a permanent DOI after peer-review.

l149 I would suggest to compare the events detected in the last ~20-30 years when using your full-record climatology and a more standard 30-years baseline.

Please see our response to your comment above regarding the choice of baseline.

l161 explain 'cumulative intensity'

We have added the following text to Section 3.3:

*"The intensity is the temperature anomaly relative to the chosen statistic (e.g. mean climatology) (Hobday et al., 2016). Over the duration of a MHW/MCS, the cumulative intensity is the sum of the daily temperature intensities, while the variance of intensity is the variation (Hobday et al., 2016). We provide event intensity variables that are relative to both the mean and threshold (e.g. 90th percentile) climatology (see Table 2). The onset rate represents the temperature change rate from the start of a MHW/MCS to the maximum intensity, while the decline rate represents the temperature change rate from the maximum intensity to the end of the MHW/MCS (Hobday et al., 2016)."*

l165 remove space between the circle sign and C

We have removed the space between the degrees symbol and 'C' throughout the manuscript.

l170 without context, Table 3 is not very useful. Please add some references.

We have added the following paragraph to the introduction:

"*There are numerous examples of extremely warm or cold temperatures being recorded in Australian coastal waters. In 2011 off Western Australia, intense low pressure linked to a strong La Niña intensified the Leeuwin current leading to MHW conditions at the surface. This MHW lasted for months and maximum intensities peaked at 5°C warmer than the 2000–2009 average climatology (Feng et al., 2013). After this event, coastal waters off Western Australia experienced further MHWs in the austral summer of following years before sea surface temperatures switched to a multi-year cold phase and when multiple MCSs occurred between 2016 and 2019 (Feng et al., 2021). In late 2015 / early 2016 off eastern Tasmania, the East Australian Current (EAC) Extension waters were warmer than average leading to MHW conditions over a large area of the Tasman Sea. The regionally-averaged MHW conditions at the surface persisted for 251 days between September 2015 and May 2016 and regionally-averaged sea surface temperature anomalies were between 1.5 and 3°C warmer than average between November 2015 and February 2016 (Oliver et al., 2017). A similar event also occurred during austral summer 2016/2017 (Kajtar et al., 2022). MHW conditions persisted for 3 months, with mean Tasman sea surface temperatures 1 to 3°C above the 1983 to 2012 seasonal climatology (Kajtar et al., 2022). Finally, in austral summer 2021/2022 the position of the EAC jet and its eddies off southeastern Australia had a crucial role in driving MHW/MCS conditions in coastal waters close to Sydney. A MHW occurred in December 2021 and February 2022, while a MCS occurred in January 2022 (Li et al., 2023).*"

We have also added some further discussion in Section 5:

"*The most intense MHWs were at approximately 22 m depth at sites PHB/PH100 and MAI in 2015 and 2015/16, respectively (Fig 4a,b). These coastal events coincide with the far-reaching MHW conditions in the Tasman Sea related to the East Australian Current (Oliver et al., 2017).*"

"*The longest MHW detected lasted 138 days at MAI (same as in Fig . 4) between 28/12/2015 and 13/05/2016 coinciding with the unprecedented Tasman Sea MHW (Oliver et al., 2017) while the longest MHWs at PHB/PH100 and ROT lasted 54 and 60 days, respectively. The longest ROT MHW occurred at the time of the `Ningaloo' MHW (Feng et al., 2013).*"

l172 and elsewhere, I think units should be 'C * day', not 'C / day'. These are the units for onset/decline rates instead. This seems correct in the netCDF files.

This was a mistake in the submitted manuscript, and we have now corrected the units for cumulative intensity to be '°C days'. The file is correct – thanks for checking.

l221 are you also using zenodo?

As explained above, we will upload the tutorials to Zenodo with a DOI at the time of publication.

l225 NetCDF is not necessarily a python format, it is actually aimed at portability.

We have removed '(python)' on L225

Please rephrase 'appendix section XX' => 'appendix XX'

We have changed the text as suggested.

---

## Author Comment (AC2)

**General Comments**

This paper presents an observational dataset documenting extreme ocean temperatures observed at four coastal stations in the vicinity of Australia, referred to as AMDOT-EXT. This dataset serves as an extension of the previously published AMDOT dataset, encompassing both surface and subsurface temperature records.

The study adheres to established standards and utilizes publicly available algorithms to identify and characterize heatwaves.

Given the global concern regarding the warming of the Earth's oceans, the dataset presented herein bears potential significance for climate impact studies and warrants consideration for publication, notwithstanding its inherent limitations. However, a substantial area in need of improvement lies in the discussion of uncertainties.

Thank you for reviewing our paper.

**Specific Comments**

The unique and pertinent attributes of this dataset, despite its limited coverage of only four stations, could be emphasized, particularly in Section 2.

We have added the following text to Section 2:

*"These sites are unique in Australia and the southern hemisphere in that they have been occupied weekly to monthly since the 1940s/50s ..."*

*"The west coast site is close to Rottnest Island (ROT, ~32°) offshore from Perth, Western Australia in approximately 55 m of water and temperatures here are influenced by the Leeuwin current. The southern site is close to Maria Island (MAI, ~42.6°), Tasmania in approximately 90 m of water and is situated in the southern extension region of the EAC. An additional two east coast sites are located close to Sydney (PHA and PHB/PH100, ~34.1°), New South Wales in approximately 50 and 100 m of water, respectively, downstream of the typical EAC separation zone in what are usually eddy dominated waters."*

*"Unlike most long-term ocean temperature data sets ,  measurements have been collected at multiple depths through the water column at PHA, PHB/PH100, MAI and ROT since the 1940/50s."*

Notably, Table 2 reveals a significant discrepancy in the "relative % record available" between station PHA and the other stations (PHB/PH100, MAI, and ROT). For PHA, data availability remains relatively uniform throughout the water column and generally falls below 100% below the surface. Conversely, for the other stations, data availability increases significantly at greater depths compared to the surface, with ROT, for instance, exhibiting 172% data availability at a depth of 29 meters relative to the surface. An explanation for this peculiar behavior would be valuable.

At PHA, we have bottle and CTD profiles only. This means that data availability is consistent throughout the water column over time at the optimal depths. At PHB, MAI, and ROT, we have a mix of bottle, CTD, mooring and satellite data sets. Percentage wise, most of the data contained within the AMDOT-EXT data products were collected after 2008/2009 when we have high-frequency (5 minute) mooring data, however most (and at ROT all) of these data are below the surface. Between 2012 and 2022 at times when we don't have in situ (mooring or CTD data) at the surface, we use satellite data. However, satellite data can be gappy (due to the impact of clouds) which means we have less data at the surface when compared with sub-surface depths.

We have added the following text in Section 3.1:

"*Data availability (Table 1 and Fig. 2) is consistent through the water column at PHA because the temperature record consists of bottle and CTD profiles only. However, at PHB/PH100, MAI, and ROT there is increased sub-surface data availability relative to the surface. This is because the satellite-derived SST data that we use starts in 2012 and has some gaps (dependent on clear skies).*"

Please note that we updated the AMDOT-EXT data products because the Australian Ocean Data Network has provided an improved SST data product, that now provides SST measurements at quality level FV02 between 2019 and 2022, instead of quality level FV01. In the process, it was noticed that the domains selected for area-averaging SST data were approximately 17 to 26 km away from the in situ locations, which has been rectified, with the SST domains now immediately encompassing the in situ locations.

This means that the results in Tables 1 and 3, and figures 2 and 4 have changed slightly, and hence have been updated. Overall, data availability at the surface has increased at PHB/PH100 (~ +15% relative to data availability in the submitted manuscript) but has decreased at MAI (~ -4%) and ROT (~ -8%). There are now more MHWs and HSs at the surface at PHB/PH100. There are also minor changes in statistics at depths below the surface at some sites because of the interpolation technique used to produce the AMDOT climatologies. We have modified the text where necessary to reflect these changes (see manuscript tracked changed document). We now also have a short MCS at MAI, which is shown in figure 2 and in figure 4 along with its characteristics.

The PHA dataset primarily comprises monthly and weekly data which is not consistent with the methodology adopted for the identification of heat waves. It is also located relatively close to the PHB/PH100. It is advisable to provide a more detailed discussion if there is a specific rationale for including this record in the dataset.

We have added the following rationale for including PHA in Section 2:

"*Although we only have bottle and CTD profiles at the PHA site at weekly to monthly sampling intervals, this site has the longest continuous temperature record in Australian waters (1942-present), as well as concurrent biogeochemical sampling. Hence, while we are not able to calculate MHWs (due to the 5 day duration required by the definition used here) we provide extreme temperature statistics at PHA as it*

*may be useful for investigating the potential impacts of extremes temperatures on biogeochemical processes."*

In reference to HS and CS (Page 6, Lines 87-88), my interpretation aligns with temperature exceeding the 90th percentile from above and the 10th percentile from below. Consequently, Table 2's columns pertaining to HSs and CSs appear non-uniform. For PHA, the number of events appears to be associated with a timescale of weeks or months (>5 days), while at the other sites, peaks are expected to encompass events lasting less than 5 days.

Thank you for bringing this to our attention. We incorrectly used the HS/CS time periods for MHW/MCSs and Vice versa. We have corrected the time periods in the updated Table 1 which will improve clarity.

The reviewer is correct that the number of HSs and CSs depend on data availability and site PHA consists of profiles collected typically weekly to monthly. The difference at the other sites is that a mooring with high frequency sampling (5 min – hourly) has also been deployed enabling the detection of MHWs and MCSs as per the Hobday et al. definition. HSs and CSs are days that exceed their temperature thresholds (90$^{th}$ and 10$^{th}$ percentiles, respectively), and do not include days counted as a MHW or MCS. We have clarified this in the edited caption for Table 1 copied below. The different data availability in time and space at the sites, as well as different ocean characteristics, explain the non-uniformity that the reviewer refers to, which we have discussed in Section 3.4.

Furthermore, it remains unclear whether "#Days Sampled" for PHS refers to the number of (weekly?) samples or the total number of days.

In Table 1 "#Days Sampled" refers to the unique number of days sampled. For example, at 29m depth at ROT 5199 unique days (or daily data points) are available between 1951 and 2022. To further clarify this, we have edited the caption for Table 1 as follows:

*Table 1: "Table showing the number of Marine Heatwaves (MHWs), Marine Cold-Spells (MCSs), Heat Spikes (HSs) and Cold Spikes (CSs) identified at sites Port Hacking A and B (PHA and PHB/PH100, respectively), Maria Island (MAI), and Rottnest Island (ROT) at multiple depths. HSs and CSs include days when temperatures were higher than the 90$^{th}$ percentile, and lower than the 10$^{th}$ percentile, respectively, and do not include days identified during MHWs and MCSs. The time period used for MHW and MCS event detection is shown separately to the time period used to detect HSs and CSs. The total record available at 2 m depth as a percentage of the time period between the first and last sampled temperatures is listed for each site, and sub-surface data availability relative to 2 m depth is shown at the other depths. The unique number of days (or daily-binned data points) available over the site's record (e.g. between 1951 and 2022 at ROT) are also shown (# Days Sampled)."*

Additionally, we now describe variables 'TEMP_HEAT_SPIKE' and 'TEMP_COLD_SPIKE' in Table 2 to avoid confusion. These variables include all temperatures that exceed the percentile thresholds (clarified in the table). However, the #HSs / CSs in Table 1 are the number of temperatures that exceed the percentile thresholds when there are no MHWs / MCSs.

While the paper acknowledges the impact of employing climatology periods of varying durations (Page 8), it does not provide an analysis of the consequences of selecting different climatology periods. Such an analysis is particularly crucial for this dataset since continuous daily records may not always be available (Figure 2), and the presence of long-term trends cannot be disregarded. In this context, the evaluation of designating the entire 80 year period as a reference climatology remains lacking. A plausible approach could involve generating different realizations of MHW and MCS to reflect data discontinuities as errors, for instance, in the count of heatwaves. Consequently, the statement asserting that the "data products highlight the challenges encountered when detecting extreme temperature events using sporadic long time series" receives weak support from the limited uncertainty analysis presented in the paper.

This is indeed an area of ongoing research, and there is much debate in the community on which way is best (e.g. Amaya et al., 2023; Sen Gupta, 2023; Rosselló et al., 2023), and we believe the answer is dependent on the question being asked. In our case we have chosen to use the full record and have been very clear about the time period used, as stated in a dedicated section 3.4 'Effects of Data Availability and Choice of Baseline'. We do this as it would be challenging to compare events using a 30-year climatology below the surface due to a lack of data at our sites.

We have edited Section 3.4 for additional clarity:

*"We acknowledge that using a climatology of approximately 30 years is recommended (Hobday et al.,2016; WMO, 2018), and is commonly used for surface MHW studies (e.g. Frölicher et al. (2018); Oliver et al. (2018); Elzahaby and Schaeffer (2019)) where such data exists. Schlegel et al. (2019) showed that using a climatology period > 30 years may affect MHW detection in a similar way as using a climatology period < 30 years, dependent on environmental multi-decadal variability. Further, Rosello et al. (2023) highlight the impact of long-term trends on MHWs and argue that a moving baseline is more adequate for detecting consistently rare extremes, although presently there is much debate on this subject (Amaya et al. (2023); Sen Gupta (2023)). We use daily climatologies calculated using the entire temperature record at the long-term sites because 30 year daily sub-surface data sets do not yet exist. At this time, high-frequency sub-surface data at the 4 sites only extend back approximately 13 to 14 years. We acknowledge that the number of extreme temperature events and their metrics will differ in the future when a 30-year daily climatology becomes available, and that it will vary depending on which 30-year period is used (as suggested by Schlegel et al 2019)."*

Amaya, Dillon J., Michael G. Jacox, Melanie R. Fewings, Vincent S. Saba, Malte F. Stuecker, Ryan R. Rykaczewski, Andrew C. Ross, et al. "Marine Heatwaves Need Clear Definitions so Coastal Communities Can Adapt." Nature 616, no. 7955 (April 2023): 29–32. https://doi.org/10.1038/d41586-023-00924-2.

Sen Gupta, Alex. "Marine Heatwaves: Definition Duel Heats Up." Nature 617, no. 7961 (May 16, 2023): 465–465. https://doi.org/10.1038/d41586-023-01619-4.

Rosselló, P., Pascual, A., & Combes, V. (2023). "Assessing marine heat waves in the Mediterranean Sea: a comparison of fixed and moving baseline methods," Frontiers in Marine Science, Volume 10, Page 1168368

Analysis quantifying the consequences of selecting different climatology periods is out of scope for this paper as our focus is to describe and explore the new AMDOT-EXT data products. This additional analysis would warrant a whole new paper, when considering work undertaken by e.g. Rosello *et al*., (2023) and Schlegel *et al*., (2019). These studies suggest that time series of different lengths will have different trends, and hence the climatologies will differ.

We have hence not included additional text on this in the manuscript.

In the context of uncertainty assessment, it is also important to account for the influence of secular trends, as exemplified in recent work on marine heatwaves in the Mediterranean by Rosselló, P., Pascual, A., & Combes, V. (2023), titled "Assessing marine heat waves in the Mediterranean Sea: a comparison of fixed and moving baseline methods," published in Frontiers in Marine Science, Volume 10, Page 1168368.

As discussed in our response above, we think that an analysis of how trends affect extreme temperature statistics is out of scope for this paper. Notwithstanding this, the data are available should the reader choose to do their own analysis. However, as mentioned we had already included discussion on this subject in a dedicated section 'Effects of Data Availability and Choice of Baseline'.

In addition to this, we have added the following text in Section 3.4:

"*Further, Rosselló et al., (2023) highlight the impact of long-term trends on MHWs and argue that a moving baseline is more adequate for detecting consistently rare extremes, although presently there is much debate on this subject (Amaya et al., 2023; Sen gupta, 2023).*"

Information regarding potential future updates to the dataset is not provided. Even the acknowledgment of an absence of clear plans for future updates constitutes valuable information.

We have added the following text in the Abstract:

"*The 4 data products are provided as CF-compliant NetCDF files and it is the intention that they be updated periodically. Please refer to the URL in the Code and Data Availability Section for the latest version.*"

**Technical Corrections**

On Page 6, Lines 87-88, for clarity, it is advisable to specify that HS and CS refer to temperature exceeding the 90th percentile from above and the 10th percentile from below.

We have updated the sentence as follows:

"*We identify HSs as temperatures that are higher than the daily-varying $90^{th}$ percentiles and CSs as temperatures that are lower than the daily-varying $10^{th}$ percentiles.*"

The final paragraph on Page 6 (Lines 102-105) is not sufficiently clear and warrants rephrasing.

We have rephrased this final paragraph as follows:

"*The AMDOT-EXT data products sometimes contain extreme events that are not consistent over depth (Fig. 3). For example, event start and end dates, and their duration, can vary as a function of depth. This is partially because data availability over time is often non-uniform through the water column, but also because the depth structure of MHW events can vary due to ocean dynamics. For example in the coastal*

*Sydney region, Schaeffer et al., 2023  showed that MHWs can be shallow, can extend across the whole water column, or can be sub-surface only. For example, during atmospherically-forced shallow MHWs we might expect there to be fewer days when temperatures exceed the 90$^{th}$ percentile at depths below the mixed layer depth when compared with temperatures at the surface (Schaeffer et al., 2023)."*

The presence of a question mark at Line 105 is likely a typographical error.

Thank you.

---

## Author Response (AR2)

In the revised version of the paper, the authors have satisfactorily addressed the issues raised in the previous round. The new figures are useful to illustrate events, but an extra step could be taken to improve accessibility.

Thank you for your time and comments. We have addressed the remaining issues below.

Table 1 is now easier to read but more explanation on the percentages should be provided. Why in some cases measurements at say 50m depth are more than those at 2m? This is counterintuitive to me. I would not consider satellite SST as the "reference", but compare the in-situ data at different depths instead.

We have changed the percentages in table 1 to be based on data availability at each depth, rather than percentages relative to the surface.

With regards to why there are more data below the surface, we had added the following text to Section 3.1:

"*Data availability (Table 1 and Fig. 2) is consistent through the water column at PHA because the temperature record consists of bottle and CTD profiles only. However, at PHB/PH100, MAI, and ROT there is increased sub-surface data availability relative to the surface (satellite). This is because the satellite-derived SST data that we use starts in 2012 and has some gaps (dependent on clear skies).*"

In the same Table 1, why MHWs and MCSs for PHA are given for a different period than other stations?

We have edited the table so that there is just one time period specified for all event types at this site, and we will work with the typesetter to remove the double-column.

Figure 2 is still hard to interpret; you could consider to add an additional figure, showing e.g. yearly mean temperatures for each station and depth (if higher frequency is not suitable).
This would support the discussion in Section 3.4 on baselines.

We have added a new figure – now Figure 3 (copied below), that shows the annual median temperature anomalies at each site and depth. These annual median temperature anomalies show an increasing trend, with earlier years generally being anomalously cold, and recent years anomalously warm. We now refer to this figure in Section 3.1:

"*The annual median temperatures for each site are shown in Figure 3.*"

and Section 3.4, along with text referencing a recent study on ocean trends at PHB/PH100 and MAI:

"*Temperatures at two of the sites: PHB/PH100 and MAI, have increased non-linearly over time, with mean temperatures (dependent on depth) in 2022 approximately 0.5 to 1.3 °C warmer relative to the 1940/50s (Hemming et al., (2023)), and annual median temperature anomalies can be seen in Figure 3.*"

[Figure]

**Figure 3.** Annual median temperature anomalies relative to the mean daily-varying climatology at optimal depths for the a) Port Hacking 50 m (PHA), b) Port Hacking 100 m (PHB/PH100), c) Maria Island (MAI), and d) Rottnest Island (ROT) sites. All years and depths shown have measurements available in at least 4 months of the year, while those years and depths with a solid black outline have measurements available in at least 10 months of the year.

Section 'Code and data availability' does not seem updated as stated in
the response to reviewers. Please avoid repetition.

> Thank you for highlighting this. We have removed the text in the 'Code and Data availability'
> section. Please note that for some reason the tracked changes in the 'Code and data availability'
> section do not appear in the tracked changes document, even after text has been edited. This
> appears to be a latex error.

Please double check the references - as pointed out below some look incorrect.

Specific comments (track changed document)
l113 kelvin symbol is K
l232 mismatched bracket
l348 typo
l396 typo
l424 citation looks corrupted
l439 as above

> We have made changes on L113 and L232 as requested by the reviewer.
> All citations have been updated using Zotero citation management instead. Hence, the citations
> highlighted by the reviewer have now been fixed.
>
> We have read through the manuscript once more and have made a few minor adjustments.
> Please see the tracked changes version of the manuscript for highlighted changes.